# Can Citizen Science Promote Flood Risk Communication?

**Wing Cheung [1] and David Feldman [2,\*]** 

[1]  Palomar College, Geography and Geographic Information Systems, San Marcos, CA 926091, USA; wcheung@palomar.edu

[2]  School of Social Ecology, University of California, Irvine, CA 92697, USA

\*  Correspondence: feldmand@uci.edu; Tel.: +1-949-824-9146; Fax: +1-949-824-8566

**Abstract:** This article explores the challenges facing citizen science as a means of joining the efforts of scientists and flood-risk affected stakeholders in motivating citizen involvement in identifying and mitigating flood risks. While citizen science harbors many advantages, including a penchant for collaborative research and the ability to motivate those affected by floods to work with scientists in elucidating and averting risk, it is not without challenges in its implementation. These include ensuring that scientists are willing to share authority with amateur citizen scientists, providing forums that encourage debate, and encouraging equal voice in developing flood risk mitigation strategies. We assess these challenges by noting the limited application of citizen science to flood-relevant problems in existing research and recommend future research in this area to meaningfully incorporate a "re-imagined" citizen science process that is based on the participatory theoretical framework. We also discuss one case study where the principles of collaboration, debate, and equal voice were put into play in an effort to apply citizen science and—in the long term—mitigate flood hazards in one set of communities.

**Keywords:** participation; citizen science; flood; risk management; risk communication

## 1. Introduction

Citizen science has been defined as "a process where concerned citizens, government agencies, industry, academia, community groups, and local institutions collaborate to monitor, track, and respond to issues of common community (environmental) concern" [1] (p. 274). It has been used to address environmental concerns such as biodiversity loss, climate change, water pollution, noise pollution, biosolids management, and, to a limited extent, flood hazards [2–7]. Researchers have suggested that citizen science can potentially trigger interest, raise awareness, and motivate greater citizen participation in flood risk management through engagement in explicit data collection [7,8].

Moreover, past research has noted citizen science's potential for "producing high quality data as well as unexpected insights and innovations" in domains such as biology, astronomy, and environmental science [4,9,10]. However, the number of flood hazard studies that have meaningfully integrated citizen science based on the participatory theoretical framework is limited. Thus, it remains unclear if the benefits of citizen science as demonstrated in other domains can be expected in flood hazard research and whether citizen science can promote the two-way communication of flood knowledge between researchers and stakeholders, which is crucial in mitigating flood risk and enhancing flood resilience [11]. The merits of two-way communication lie in researchers better understanding lay citizens' concerns about flood hazards and what is needed to address them as well as lay audiences engaging with researchers to monitor evolving problems and provide feedback on the perceived effectiveness of solutions. By analyzing existing citizen science activities through Fiorino's (1990)

"participatory theoretical framework", we highlight challenges encountered by the current citizen science process. We also develop recommendations for future research to meaningfully integrate citizen science in risk communication efforts. Finally, we discuss one case study implemented by the authors to reduce flood risk by better incorporating citizen-level concerns and citizen group prescriptions in flood hazard mapping tools. The entire methodology is depicted in Figure 1.

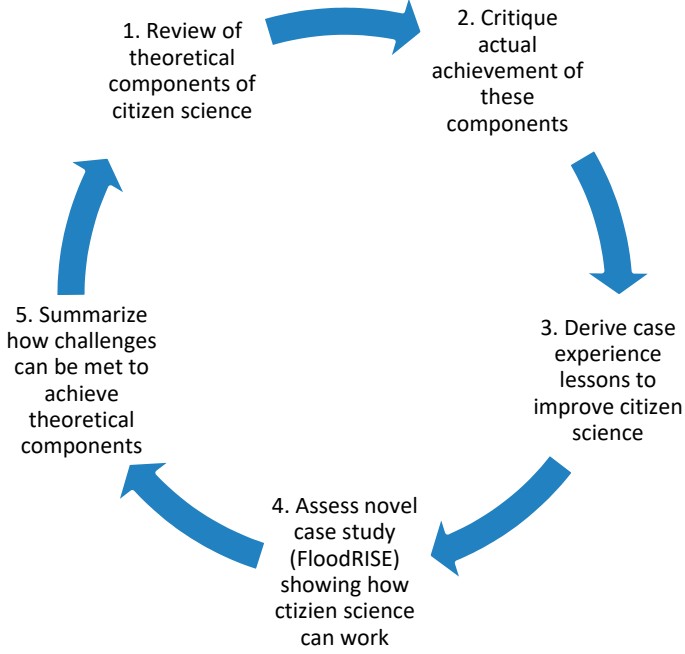

**Figure 1.** Methodology of study.

## 2. Materials and Methods

In order to evaluate the prospects for citizen science as a vehicle for enhancing citizen participation in flood risk management, we employed five basic methods, which are elaborated in Figure 1. First, we proceeded to explore the theoretical components common to such participation strategies as identified in the literature. Our goal was to identify key expectations thought to be essential to the effective operation of citizen participation initiatives in hazards or other forms of resource management related problems. Fiorino's (1990) analysis of citizen participation in environmental risk assessment was employed for this overall exploration, since his participatory theoretical framework is one of the strongest tools in the literature for providing guidance in evaluating the effectiveness of such strategies. Specifically, Fiorino's framework contends that effectiveness should be measured according to the criteria of: (1) participation by amateurs, (2) authority sharing, (3) debate and discussion, and (4) equality with officials and experts evident in each strategy found in a given initiative [12].

Second, we then proceeded to critique—given actual experiences—the challenges encountered in achieving high scores on these criteria. To accomplish this, our third step was to analyze past citizen science activities related to flood-risk management based on these four criteria in order to identify shortcomings in the current citizen science process. We further derived from these experiences an improved citizen engagement model that would involve the public in multiple stages of hazard mitigation, account for diverse stakeholder needs, and clearly set out expectations for how citizen input can be utilized in managing hazards.

Fourth, we assessed the lessons of a case study in which the authors were engaged as principal investigators to illustrate how the process of citizen science can improve hazard management tools while fulfilling community needs for education, information, and hazard visualization. This case study—called "FloodRISE", funded by the U.S. National Science Foundation—was designed from the outset to incorporate a citizen science component that would combine the resources of experts

(who developed visualization tools to depict flood hazard) with the preferences and the needs of community members (whose aspirations centered around making such tools useful and useable). This particular case was selected because of the authors' direct involvement as principal investigators, the case's important findings vis a vis citizen science outcomes, the fact that the case aptly revealed how to overcome the challenges identified in the literature, and because the case is replicable in coastal communities worldwide. The FloodRISE research team investigated, developed flood hazard maps, and initiated a citizen science process for the study's field sites in Southern California and Northern Mexico. We had the benefit of supportive local researchers in these communities who were also co-investigators on the project. Fifth and finally, we consider how citizen science challenges may be overcome given the lessons derived from the FloodRISE case.

## 3. Literature Review—Challenges in Doing Citizen Science

In order to realize the full potential of citizen science, researchers often enlist members of the public to help them gather vast amounts of data across an array of locations over the span of several years [13]. Most participants in these projects are amateurs from diverse locations and sociopolitical backgrounds and are commonly motivated to participate based on a shared sense of personal curiosity [14].

Examples of such citizen science projects share in common a promise—that of providing "underrepresented groups the opportunity to participate in a component of decision making and planning from which they might have traditionally been excluded" [15] (p. 310). The need to include diverse perspectives is demonstrated in collaborative activities such as the Southwest drought-climate variability and water management plan, where partnerships among diverse stakeholders (water managers, forecasters, and researchers), improved streamflow forecasting, and enhanced use of the scientific products by the users are all combined into a comprehensive strategy [16]. The need for such a comprehensive approach has been echoed by the findings of flood risk communication research conducted across Europe, where researchers have been urged to consider the diverse situational, social, cultural, and psychological factors among stakeholders that are believed to shape their motivation to participate in informal flood management activities such as citizen science [17].

### 3.1. Authority Sharing

In the context of Arnstein's (1969) Ladder of Citizen Participation (see Figure 2), most existing citizen science projects operate somewhere between the therapy (level 2) and the consultation (level 4) rungs of the "ladder." [18] In this top-down model, scientists and researchers often define the problems to be examined and the structure of the project with minimal input from participants [19–21]. This approach is not surprising given that many experts still subscribe to the so-called deficit model, which is based on the premise that laypeople without scientific backgrounds or training are ignorant and thus need information provided by experts to overcome their ignorance [22–25].

In this traditional top-down model, scientists usually aggregate and analyze observations collected by citizens and then present their assessment of risks and recommendations to stakeholders. Although this one-way "loading dock" model of science communication has often been the norm within the scientific community, research over the past decade has shown that more and better information by itself does not necessarily motivate flood risk mitigation or promote flood communication [26–28]. This is partly because, under the top-down model of citizen science, risk communication and definition of risk are largely "undermined by conscious or unconscious efforts" [20] to conform to deliberation to one (i.e., the scientists') particular discourse, understanding, or disciplinary perspective [11].

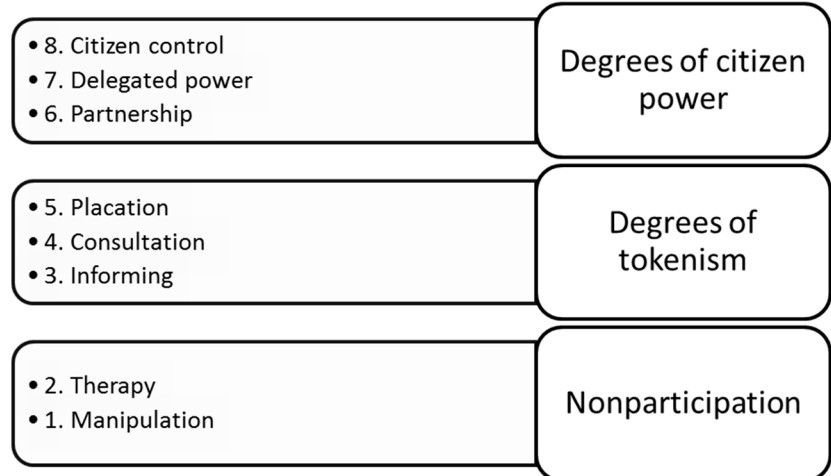

**Figure 2.** Eight levels of participation within Arnstein (1969)'s Ladder of Citizen Participation [15].

### 3.2. Debate and Discussion

Although some citizen science projects offer experienced and trained volunteers the opportunity to move beyond the bottom rungs of Arnstein's (1969) ladder of participation by soliciting participants' concerns and feedback, the flow of information is nonetheless mostly unidirectional [7]. For example, in a comparative analysis of the role of citizen observatories in flood risk management in the United Kingdom, the Netherlands, and Italy, researchers found that citizens were mostly treated as "spectators" and "human sensors" responsible for little more than providing authorities with local flood information [7]. Although feedback and discussion between scientists and citizen science practitioners (herein called "citizen scientists") have been shown to increase retention of knowledge and participation, expose myths and stereotypes, improve participant knowledge, and empower participants, projects characterized by this two-directional flow of information are the exception rather than the norm [7,13,29,30].

In fact, Rotman (2012) found that, due to a combination of apprehension and mistrust, participants are often intimidated by the scientific jargon employed in communication efforts, thus they are reluctant to present their ideas to researchers gathered together for discussions [11]. Furthermore, scientists who are understandably worried about volunteers' level of commitment and quality were at times perceived by participants as arrogant or "too concerned with having each element of the scientific study as perfect as possible" [31] (p. 34). As a result, debate and discussion, or, for that matter, any form of communication between scientists and participants, are often minimal despite the demonstrated benefits of ongoing two-way communication and feedback.

### 3.3. Participant Equality with Officials and Experts

Given the hierarchical power structure of most scientist–public exchanges as well as the lack of interaction between experts and participants in flood risk management activities, as discussed in previous sections of this paper, it follows that participants are not often treated as the equal of experts or decision-makers in formulating strategies for flood risk communication. The top-down framework of many citizen science projects may only reinforce the traditional "pipeline" notion of scientific products, which treats the public as "storage tanks" to be loaded-up with data [26]. Unfortunately, this data may be in a form that is difficult to communicate to laypersons and/or of minimal interest to affected stakeholders, at least in the view of some [26].

This differential power structure is partly reinforced by the neo-positivist character of our society and the ways it views social science research. This over-arching view "emphasize(s) empirical research designs, sampling and data gathering, measuring outcomes, and causal model with predictive power" [28]. Moreover, the disproportionate significance that public officials attach to quantitative

over qualitative information (e.g., the prospective height of flood waters likely from a storm event as opposed to the perceived amenity value of some critical habitat likely to become flooded, even if the latter has little economic value) when using science for decision-making further entrenches a preference for technical expertise over experiential knowledge [21,22].

In short, these top-down and technocratic approaches clearly fail to account for the local and the experiential knowledge that is often only accessible to local communities themselves. These sources of knowledge are crucial for furthering the understanding of the wider political, economic, and legal-regulatory settings that determine the acceptability of a new technology or a proposed course of action for the management of risk [23,32].

*3.4. Re-Imagining Citizen Science*

As demonstrated in the vast majority of flood-related citizen science projects and other citizen engagement activities that we reviewed, citizen science has the potential to provide flood managers and flood-risk researchers with vast amounts of data and unexpected insights while empowering participants and motivating their participation in flood risk management activities [7,10,32]. However, these positive outcomes can only be fully realized in a meaningful participation process if the research is viewed as legitimate, has a clear and discernible purpose, and genuinely involves stakeholders in its generation and dissemination [33]. This vision of a re-imagined citizen science in flood communication is supported by Korfmacher (2001), who similarly called for a public participation process that: (1) is transparent and easy to understand, (2) involves the public in multiples stages of development, (3) accounts for the preferences of diverse stakeholders (organized and unorganized) and provides them with regular feedback, (4) contains mechanisms for resolving disputes between non-experts and experts, and (5) sets expectations as to how project outcomes will be used in decision-making or research [34].

## 4. Findings and Lessons from an Empirical Case

Based on Korfmacher's (2001) recommendations, the current process of citizen science could be "re-imagined" (perhaps re-designed or re-engineered is a better way of describing his claims) by complementing the top-down structure of existing citizen science projects with a bottom-up approach that can account for various social, cultural, and contextual factors in the communication of flood risk [35]. In past studies of public behavioral responses to floods, it has been found that the top-down, official mode of communicating risk fails for a variety of reasons (e.g., patchy warning, short lead time, forecast error, poor warning dissemination) [7,17]. Instead, Parker et al. (2009) found that non-formal activities such as citizen workshops, learning-by-doing activities, and engaging stakeholders in flood risk management activities are keys to communicating flood risk information and building community resilience [17].

A robust sample of these types of activities was performed by the Flood Resilient Infrastructure and Sustainable Environments (FloodRISE) project at the University of California Irvine, where teams of interdisciplinary researchers co-produced flood hazard maps and geographic information system tools by engaging diverse stakeholder groups in a series of surveys, focus groups, as well as training and outreach workshops [35].

While household surveys and mapping exercises demonstrated the extent to which local residents were aware of local flood risk, the focus groups organized in flood-vulnerable coastal communities in Orange County, California and Tijuana, Mexico were utilized to compare and evaluate a series of flood hazard maps generated by researchers. Composed of emergency managers, emergency responders, local officials, and citizen activists, two recurring themes that emerged from these focus group discussions are that (1) different users have different needs with regard to flood maps, and (2) these requirements might not necessarily be in accord with nor fully understood by map producers. The substance of these focus group discussions—specifically the call for more user-friendly

maps—resonates with previous research findings, which have called for a map creation process that incorporates meaningful public participation in an effort to identify user requirements [36].

The FloodRISE researchers also discovered that decision-makers have divergent "decisional spaces", which render the coordination of disaster responses and the setting of emergency priorities especially challenging. Moreover, some decision-makers—such as emergency managers, for instance—have little time in which to make critical decisions and operate under pressure from political, legal, economic, and other community interests [37]. In short, what the research team ultimately learned from this citizen science exercise is that making risk-reduction tools such as flood maps more useful requires more than instructing citizens about risk; it requires directly involving lay citizens and experts in two-way communication to fully identify the range of risks relevant to different groups, to enhance the value of tools in mitigating those risks, and to ensure their usefulness to the people who will implement them to mitigate hazards.

A further discovery from the focus groups was that, because no single map could fully address a single flood-related issue, participants in virtually every case appreciated the employment of various flood risk representations. These representations depicted different types of vulnerable resources even though only a limited number of scenarios may be relevant to their decisional space. In sum, we found that decision-makers can benefit from a variety of flood risk representations that can be likened to a "menu" of visualization tools. This is because diverse publics value different maps based on their roles and responsibilities, and flood risk maps serve a variety of personal and professional needs. In effect, focus groups are a means of generating two-way communication between researchers and citizens regarding flood hazards.

While visualization tools may illuminate flood hazards, interactive exchanges between map-creators and decision-makers allow for a sharing of policy-specific knowledge regarding flood vulnerabilities from the vantage point of communities. Thus, information discerned from the focus groups can be used to refine the visualization tools themselves. Moreover, the focus group structure acknowledges that this community-level knowledge differs from locality to locality.

Taking these findings one step further and incorporating its implications into policy, Paul et al. (2018) recommends that citizen science should shift from a "citizen sensor" model to one where citizens are not only performing flood monitoring tasks but are actually involved in the entire project's life-cycle. Specifically, citizens should be involved in setting the scientific agenda, helping determine what information is needed and how it might be gathered, and monitoring its acquisition [33]. In sum, while there is still a need for officials to fund or coordinate hazards research, experience has shown that the top-down approach to hazard management should perhaps be complemented by non-formal learning approaches facilitated in part by a bottom-up model of citizen science [17].

In addition, since it is also true that citizen participants and experts both contribute in different ways to citizen science projects, "scientists need to develop respect towards those who help them beyond the realization that they provide free labour" [19] (p. 118). Specifically, previous studies as well as risk communication theories have shown that "the public has useful knowledge and concerns that need to be acknowledged", and joint fact-finding activities such as citizen science can generate research outcomes that are more useful to policymakers and more credible to the affected public [6]. By being respectful of stakeholders' divergent values and beliefs, the public is likely to perceive the research effort and its outcome as legitimate [6].

This claim is also supported by the guidelines for the development of optimal decision support systems for risks from climate variability, which similarly stress the need to involve disadvantaged populations in order to address the inequitable distribution of information and participation opportunity [16]. Moreover, the objectives of the project should be ideally agreed upon based on a consensus between experts and diverse stakeholder groups since the impacts of such projects may have disproportionate impacts on environmentally-disadvantaged groups, as shown, for example, by Hurricane Katrina.

However, since this level of involvement may not always be possible due to constraints of funding and time, researchers should announce at the outset the limitations (including uncertainties) and the objectives of the citizen science project. This is a concern experienced even by successful citizen science projects such as the Neighborhood Nestwatch project, where 44% of its participants did not understand the overall goals of the project and "expressed reasonable concerns about the quantity and quality of the data [29] (p. 591)."

## 5. Conclusions

Citizen science approaches offer great potential for raising flood awareness, empowering local populations, and building community resilience against the hazards of flooding to life and property. When implemented in a meaningful and proactive fashion—as the FloodRISE project demonstrated—citizen science is likely to harbor comparable potential for promoting flood risk communication despite its thus far limited use in flood hazard mitigation research. By re-imagining the structure of citizen science as a two-way communication model between researchers and stakeholders, and one that invites diverse stakeholder groups to participate in the entire project cycle, it can perhaps help to avoid "the distrust felt by citizens who complain that public participation programs are a charade" [38] (p. 462).

FloodRISE found that citizen science can help change how experts and non-experts can engage each other as they seek to reduce the impacts of flooding and build more resilient communities. Essentially, the principal investigators found that, while engineers and social scientists may be able to design good tools to depict flood hazards (e.g., precise maps), citizen engagement enables scientists to collaborate with non-experts to co-produce relevant flood hazard mapping tools that are responsive to user needs. Given that no specific group of experts or non-experts has complete knowledge of flood risk and vulnerability, such a collaborative framework helps ensure that communication and management strategies will provide actionable and credible information, as shown in the FloodRISE project.

Citizen science can also promote itself as a mechanism for communicating risk and enhancing deliberation among diverse stakeholder groups affected by flooding as a natural hazard [13,34]. As we confront an era of increasing flood risk, particularly to coastal communities throughout the world as a consequence of climate change, such two-way communication and stakeholder-science partnership will become even more critical.

**Author Contributions:** Conceptualization, W.C. and D.F.; Methodology, W.C.; Lessons and Findings, D.F. and W.C.

**Funding:** This research received no external funding.

**Acknowledgments:** We would like to acknowledge previous support from the National Science Foundation under Grant No 1331611 (the FloodRISE project, which ended in 2017). The authors will also like to thank survey participants, focus group participants, and the research team which whom we worked in this previous project.

**Conflicts of Interest:** The authors declare no conflict of interest.

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
