# Peer review of "Can Citizen Science Promote Flood Risk Communication?"

_water, doi:10.3390/w11101961_

Round 1

Reviewer 1 Report

The article is reviewed according to the indications sent.

Author Response

We have completely rewritten the methods section - and the new text which describes the approaches taken on our research - are described as follows: 

In order to evaluate the prospects for citizen science as a vehicle for enhancing citizen participation in flood risk management, we employed five basic methods, which are elaborated in Figure 1. First, we proceeded to explore the theoretical components common to such participation strategies as identified in the literature. Our goal was to identify key expectations thought to be essential to the effective operation of citizen participation initiatives in hazards or other forms of resource management related problems. Fiorino’s (1990) analysis of citizen participation in environmental risk assessment was employed for this overall exploration, since his participatory theoretical framework is one of the strongest tools in the literature for providing guidance in evaluating the effectiveness of such strategies. Specifically, Fiorino’s framework contends that effectiveness should be measured according to the criteria of: (1) participation by amateurs, (2) authority sharing, (3) debate and discussion, and (4) equality with officials and experts evident in each strategy found in a given initiative [12].
Second, we then proceeded to critique – given actual experiences – the challenges encountered in achieving high scores on these criteria. To accomplish this, our third step was to analyze past citizen science activities related to flood-risk management based on these four criteria in order to identify shortcomings in the current citizen science process. We further derived from these experiences an improved citizen engagement model that would involve the public in multiple stages of hazard mitigation, account for diverse stakeholder needs, and clearly set out expectations for how citizen input can be utilized in managing hazards.
Fourth, we assessed the lessons of a case study in which the authors were engaged as principal investigators to illustrate how the process of citizen science can improve hazard management tools while fulfilling community needs for education, information, and hazard visualization. This case study – called “FloodRISE,” funded by the U.S. National Science Foundation– was designed from the outset to incorporate a citizen science component that would combine the resources of experts (who developed visualization tools to depict flood hazard) with the preferences and needs of community members (whose aspirations centered around making such tools useful and useable). This particular case was selected because of the authors’ direct involvement as principal investigators; the case’s important findings vis a vis citizen science outcomes; the fact that the case aptly revealed how to overcome the challenges identified in the literature; and because the case is replicable in coastal communities worldwide. The FloodRISE research team investigated, developed flood hazard maps, and initiated a citizen science process for the study’s field sites in Southern California and Northern Mexico. We had the benefit of supportive local researchers in these communities who were also co-investigators on the project. Fifth and finally, we consider how citizen science challenges may be overcome given the lessons derived from the FloodRISE case.

Reviewer 2 Report

This is an interesting topic discussed. Citizen science is an emerging field that is expected to grow in the near future.

The authors pose an interesting question in the title and discuss interesting and significant works in the literature in an effort to provide an answer.

Their research presentation seems initially effective, however at the last parts of the manuscript feels like you are not convinced by their arguments. The case study that they are discussing probably can be exploited or explained in more detail to convince the reader that indeed, in practice, they measured or found  that the bottom up approach is working better.

The manuscript would be greatly benefit from such an addition.

Overall, the manuscript at its current form, mostly discusses the literature and then form and describes an opinion. I would argue that I would benefit from presentation of some form of findings that convince the reader 1) that citizen science has a merit in risk comm and b) that this study has an added value , presenting something new, rather than reviewing previous works.

I suggest that the manuscript should be considered for publication, but with major revisions.

Specific comments:

line 32: explicit data collection is indeed what is being done. The authors should include recent literature for example: Assumpção et al (2019) Citizens’ campaigns for environmental water monitoring: lessons from field experiments, IEEE Access PP(99):1-1 DOI: 10.1109/ACCESS.2019.2939471

line 38: the authors could explain perhaps a little bit better the merit of the two-way communication.

line 41: please explain the term participatory theoretical framework in simple terms

lines 39-41: I think sentence is too long. The reader would benefit if you could simplify it or break it in two.

line 48: The chapter would benefit from a small diagram presenting in brief the methods followed.

line 53-54: same as above. Please simplify sentence.

line 84: The title of chapter 3. Since this chapter discusses other studies findings and conclusions I doubt “results” is the best title. It would be better to choose something like “literature review”, “current research findings” or something like that.

line 138: IT would be good for the manuscript to show how many of the studies (or the significant studies) or application of citizen science in the field, are using the two-way communication out of the total. Is the majority one-way or two way? This would strengthen the paper, because it would show a quantitative aspect of the analysis, which is not apparent in its current form.

lines 148-150: this is kind of an opinion, or it comes out like that. It is ok to include it, but this chapter is supposed to present facts / findings. Opinions could fit better in the discussion or conclusion chapters.

lines 152-153: Can you give an example of quantitative and qualitative information, so the difference would be more obvious to the reader?

line 174: I would not call this chapter “discussion”. Since it examines a case study, it would be very beneficial for the manuscript to analyse it and present the findings, rather than discussion.

line 191: please correct the spelling in property and flooding.

195: “these focus group discussions showed”. Where this is shown? It is important to clarify and present why this is shown, because in the next few lines the authors use this claim to support their conclusions. It would be good if more details are included in this manuscript regarding the case study, as they can convince the reader about the merits of the shared map development etc.

line 197: “this finding lends support….”. To convince the reader for this argument, the authors should say more details about the “finding”.

lines 240-242: Conclusions should not incorporate references to support claims. Conclusions should be based on the study’s results. In general the chapter should discuss what the authors found rather than discussing how their opinions fit with others findings.

 line 304: link for Parker et al. is not working I think. Please check it again.

Author Response

We thank Reviewer #2 for the comments provided, and respond to them as follows. Please note that because of additional text, and editorial changes, lines in the manuscript may not align perfectly with the lines in the previous manuscript. We have, however, responded to each of the comments and made changes accordingly, as noted below.

line 32: We have added a reference to: Assumpção, Thaine Herman, Andreja Jonoski, Iouliani, Theda, Ioana Popescu. “Citizens’ campaigns for environmental water monitoring: lessons from field experiments, September 2019 IEEE Access (99):1-1.DOI: 10.1109/ACCESS.2019.2939471

line 38(9): on the merits of two-way communication, we now state: "The merits of two-way communication lie in researchers better understanding lay citizens’ concerns with respect to identifying hazards and what’s needed to address them, and lay audiences engaging with scientists in monitoring evolving problems and providing feedback on the perceived effectiveness of solutions."

line 41: the phrase "participatory theoretical framework" has been replaced with "citizen science activities." 

lines 39-41: sentences have been broken up and are now more readable.

line 48: a new Figure 1 has been provided at the end of the paper presenting in brief the manuscript's methodology.

line 53-54: again, sentences have been shortened/condensed and made more readable.

line 84(86): Chapter 3 has been appropriately re-named: "Literature review – challenges in doing citizen science"

line 138: Now the claim is stated as: "Given the hierarchical power structure of most scientist–public exchanges, as well as the lack of interaction between experts and participants in flood risk management activities, as discussed in previous sections of this paper...."

lines 148-150: We have rephrased the sentence to qualify the fact that while an opinion in the literature, it has resonance in the findings of our paper. We feel this is the most appropriate way to go.

lines 152-3: examples of quantitative/qualitative information, to wit: "Moreover, the disproportionate significance which public officials attach to quantitative over qualitative information (e.g., the prospective height of flood waters likely from a storm event, as opposed to the perceived amenity value of some critical habitat likely to become flooded, even if the latter has little economic value) when using science for decision-making further entrenches a preference for technical expertise over experiential knowledge."

line 174: Chapter 4 has been appropriately re-named: "Findings and lessons from an empirical case"

line 191: spelling has been corrected for "property" and 'flooding"

line 195: sentence re-phrased as follows: "...these focus group discussions provided evidence to the principal investigators who were present and able to observe their interactions that different users have different needs with regard to flood maps, and that these requirements might not necessarily be in accord with, nor fully understood by, map producers.

line 197(201): sentence re-phrased as follows: "The substance of these focus group discussions – specifically the call for more user-friendly maps – resonates with previous research findings, which have called for a map creation process that incorporates meaningful public participation in an effort to identify user requirements." 

lines 215-228: we added considerable new material to the conclusion; to wit: "A further discovery from the focus groups was that because no single map could fully address a single flood-related issue, participants in virtually every case appreciated the employment of various flood risk representations. These representations depicted different types of vulnerable resources, even though only a limited number of scenarios may be relevant to their decisional space. In sum, we found that decision-makers can benefit from a variety of flood risk representations that can be likened to a “menu” of visualization tools. This is because diverse publics value different maps based on their roles and responsibilities, and flood risk maps serve a variety of personal and professional needs. In effect, focus groups are a means of generating two-way communication between researchers and citizens regarding flood hazards.
While visualization tools may illuminate flood hazards, interactive exchanges between map-creators and decision-makers allow for a sharing of policy-specific knowledge regarding flood vulnerabilities from the vantage point of communities. Thus, information discerned from the focus groups can be used to refine the visualization tools themselves. Moreover, the focus group structure acknowledges that this community-level knowledge differs from locality to locality."

line 335: the Parker reference link has been replaced with a new link that now works.

Round 2

Reviewer 2 Report

The revised version is very improved in comparison to the original manuscript. The authors did a very good job in improving small details in the manuscript that gave an overall greatly improved result. They perfectly understand my concerns in the initial review and addressed all the deficiencies that according to my opinion were present in the original manuscript. I suggest that the manuscript can be accepted for publication in its current form.

This manuscript is a resubmission of an earlier submission. The following is a list of the peer review reports and author responses from that submission.

Round 1

Reviewer 1 Report

The relationship between local knowledge and citizen science and contributing to the formation of flood-resilient communities should be more grounded.

Reviewer 2 Report

This paper provides an interesting discussion of the literature on citizen science and how scientist/lay communication could become a central component of citizen science projects. The paper is well written and the topic is of interest to readers of Water.

The key problem, however, is that it is not clear what contribution the paper is making to the current state of knowledge. The paper does not appear to be based on a study or field experiment. Alternatively it is not clear whether the paper is based on a systematic review of the literature either. It would be useful to articulate how this paper extends the field, how the new insights were developed and how this is an advance on what has been previously published. 

Minor point - the assertion about the 'neo positivist character of our society' should be supported by evidence or tempered - which society? which groups?